# Drug Survival and Clinical Course of Patients with Cancer Treated with Biologic Therapy for Psoriasis

**DOI:** 10.3390/jcm13216546

**Published:** 2024-10-31

**Authors:** Nicole Macagno, Luca Mastorino, Michela Ortoncelli, Silvia Borriello, Chiara Astrua, Anna Verrone, Elena Stroppiana, Paolo Dapavo, Niccolò Siliquini, Simone Ribero, Pietro Quaglino

**Affiliations:** Dermatology Clinic, Medical Sciences Department, University of Turin, 10123 Torino, Italy; nicole.macagno@edu.unito.it (N.M.); mortoncelli@cittadellasalute.to.it (M.O.); silvia.borriello@edu.unito.it (S.B.); castrua@cittadellasalute.to.it (C.A.); averrone@cittadellasalute.to.it (A.V.); estroppiana@cittadellasalute.to.it (E.S.); paolo.dapavo@gmail.com (P.D.); nic.siliquini@gmail.com (N.S.); simone.ribero@unito.it (S.R.); pietro.quaglino@unito.it (P.Q.)

**Keywords:** psoriasis, PASI, cancer, oncology, safety, tumor, biologics

## Abstract

**Background/Objectives:** Patients with treated solid tumors (TST) are a highly heterogeneous and difficult-to-treat population due to the risk of disease progression/recurrence or infection. **Methods:** We conducted an observational, retrospective, single-center study at the Dermatology Clinic of Turin with a focus on the special population of cancer patients with psoriasis treated with biologics. **Results:** As of July 2023, 52 psoriatic patients with a prior/concomitant history of malignancy had taken biologic drugs. The median age was 67 years, and the median age of cancer onset was 55 years. The most common tumors were gastrointestinal cancer and melanoma. After the tumor diagnosis, 61% received an anti-IL17 drug; 37 patients continued the initiated biologic therapy, while 12 switched drugs due to secondary inefficacy. The estimated biologic DS was 55.6% at 50 months. Evidence suggests that IL-17 is a key pathogenic factor involved in tumorigenesis, resulting in a lower risk of malignancies in subjects managed with IL-17 inhibitors. Similarly, IL-23 plays a role in suppressing innate immunity and promoting tumor and metastases development. This is a consistent real-life case series that support the use of biologic drugs in patients with TST. **Conclusions**: IL-23 and IL-17 inhibitors, being immunomodulators rather than immunosuppressants, may be a safe option for patients in an active oncological setting and for immune-correlated adverse events.

## 1. Introduction

Psoriasis is a chronic inflammatory disease that affects about 2% of the general population. The number of patients with psoriasis and cancer diseases is increasing, likely due to the presence of common risk factors including smoking and alcohol consumption [1]. Moreover, interleukin (IL)-17 and IL-23 are involved in the immunopathogenesis of psoriasis and related comorbidities by acting to stimulate keratinocyte hyperproliferation and feed-forwarding circuits of perpetual T cell-mediated inflammation. In addition, the development of new psoriasis or a worsening of a pre-existing form can be secondary to immunotherapeutic therapies undertaken to treat the neoplasm [2].

Biologics, or biological drugs, are targeted therapies derived from living organisms that modulate the immune system. They can be categorized based on their specific mechanisms of action as anti-TNFa, anti-IL 23, anti-IL-12/23, and anti-IL-17. Anti-TNF-alpha agents, such as adalimumab and etanercept, inhibit tumor necrosis factor-alpha, a pro-inflammatory cytokine involved in the pathogenesis of psoriasis. Anti-IL-23 biologics like risankizumab, guselkumab, and tildrakizumab block interleukin-23, a cytokine crucial to the inflammatory process of the disease. Additionally, ustekinumab also targets both interleukin-12 and interleukin-23, helping to modulate immune responses. Anti-IL-17 therapies, including secukinumab, brodalumab, ixekizumab, and bimekizumab, inhibit interleukin-17, a cytokine that promotes inflammation and keratinocyte proliferation. Each class of biologics offers a unique mechanism of action, providing tailored treatment options for patients with moderate to severe psoriasis.

Patients with treated solid tumors (TST) are a highly heterogeneous population in terms of tumor type, stage, tumor mutational status, prognosis, and treatment received. Systemic therapies for psoriasis are not widely used in these patients because of concerns about the possibility of increased tumor recurrence/progression or infectious complications. Additionally, there is a concern that poor outcomes in cancer survivors might be ascribed to systemic therapy for psoriasis. The evidence is scarce because cancer patients are excluded from clinical trials of agents to treat psoriasis. Case series, case reports, and observational cohort studies are available, but overall, patients with TST are too few to provide meaningful results [3,4]. 

## 2. Materials and Methods

We conducted an observational, retrospective, single-center study at the Dermatology Clinic of Turin University Hospital with a focus on the special population of cancer patients with psoriasis who required biologic treatment. We collected data from patients with moderate-to-severe psoriasis treated with any biologic drug and with a prior or concomitant history of malignancy.

The aim of this paper is to describe the characteristics of this population and to evaluate the efficacy, safety, and biologic drug survival in the treatment of cancer patients with concomitant psoriasis.

Drug survival (DS) was calculated in months and was defined as the time from initiation to discontinuation (stop/switch) of the biologic treatment. The date of discontinuation was defined as the date when the treatment was interrupted for various reasons, or the date when the treatment was switched or swapped to another medication for any reason (primary inefficacy, secondary inefficacy, other).

Descriptive statistics are given as frequencies (n) associated with percentages for categorical variables and means with standard deviations (SD) for continuous variables. Survival analysis was performed using a Kaplan–Meier estimator to obtain descriptive survival curves.

The selected biologic drug, as well as factors including gender, BMI, psoriatic arthritis (PsA), diabetes, naïve to systemic therapies, and naïve to biologic therapies, were included as covariates in the adjusted model. The data were analyzed using Stata. A *p*-level of 0.05 was considered significant in all analyses.

All patients gave written informed consent. The present study was conducted in accordance with the Declaration of Helsinki initially published in 1964 on Ethical Principles for Medical Research Involving Human Subjects and approved by the local ethical committee (www.wma.net (accessed on 25 August 2024)).

## 3. Results

As of July 2023, 52 patients with a prior or concomitant history of malignancy had taken biologic drugs for the treatment of moderate-to-severe psoriasis. Of the 52 patients, 31 were male (60%) and 21 were female (40%). Regarding smoking habits, 24 patients had never smoked (46%), 16 had previously smoked (31%), and 12 actively smoked (23%). Twenty-six of the fifty-two patients had cardiovascular comorbidities (50%), nine had diabetes mellitus (17%), and ten were obese (19%).

The median age of these patients was 67 years (min: 40; max: 90). The median age of onset of psoriasis was 47 years (range: 4–80). In total, 41 patients (78%) had psoriasis vulgaris, 4 (8%) had guttate psoriasis, 4 (8%) had inverse psoriasis, 2 (4%) had psoriatic erythroderma, and 1 (2%) had pustular psoriasis. Furthermore, in 42 of the 52 patients (80%) the psoriasis affected difficult sites (scalp, palms and soles, genitals, nails). In 20 patients (38%), skin involvement was also associated with PsA.

The mean age of cancer onset was 55 years (range 2–77). Two patients had pediatric tumors. One patient had both melanoma and colorectal adenocarcinoma, and one had both bladder carcinoma and prostate carcinoma. The most common tumors were gastrointestinal cancer (*n* = 7), melanoma (*n* = 7), breast cancer (*n* = 6), and multiple non-melanoma skin-cancer (*n* = 6), as detailed in Table 1.

Additionally, 49 of the 52 patients (94%) had the tumor before starting biologic therapy for psoriasis. The mean time between diagnosis of cancer and the start of biologic therapy for psoriasis was 9.84 years (SD: 10.06), and the median time was 6.53 (range: 2–14). In these cases, after the diagnosis of the tumor, 30 patients (61%) received an anti-IL17 drug: 20 patients (41%) received secukinumab, 7 patients (14%) received brodalumab, and 3 patients (6%) received ixekizumab. A total of 20 patients (41%) received an anti-IL23 agent: 8 patients received risankizumab (16%), 7 patients received guselkumab (14%), and 5 patients received tildrakizumab (10%). Additionally, 1 patient (2%) received an anti-IL 12/23 drug (ustekinumab), and 1 patient (2%) received an anti-TNFa drug (adalimumab).

Thirty-seven of the forty-two patients continued the initiated biologic therapy, while twelve switched drugs due to secondary inefficacy: one patient with adalimumab switched to tildrakizumab (bone cancer), one on ixekizumab switched to risankizumab (gastro-enteric cancer), two on secukinumab switched to ixekizumab (melanoma, bone cancer), two on secukinumab switched to risankizumab (gastrointestinal cancer, bladder cancer), one on secukinumab switched to guselkumab and ixekizumab (testicular cancer), one on secukinumab switched to brodalumab (adrenal and pituitary tumor), two on brodalumab switched to risankizumab (bladder cancer, bone cancer), one on ustekinumab switched to secukinumab and brodalumab (gastrointestinal cancer), and one on tildrakizumab switched to guselkumab (renal tumor). The mean time of treatment with first-line biologic therapy prior to drug discontinuation and subsequent transition to second-line therapy was 22.24 months, with a range of 5.6 to 46.17 months.

As of July 2023, 28 patients (54%) are receiving an anti-IL17 drug, and 24 patients (46%) are receiving an anti-IL23 drug. Specifically, 15 patients (29%) are receiving secukinumab, 12 are receiving risankizumab (23%), 8 are receiving brodalumab (15%), 8 are receiving guselkumab (15%), 5 are receiving ixekizumab (10%), and 4 are receiving tildrakizumab (8%). One patient treated with ixekizumab is also taking omalizumab (Figure 1).

Regarding previous tumor treatment, most of the patients (37 patients, 71.5%) underwent radical surgery, 2 patients (3.8%) underwent chemotherapy, 2 patients (3.8%) underwent radiotherapy, 2 patients (3.8%) underwent surgery + chemotherapy, 3 patients (5.7%) underwent surgery + chemotherapy + radiotherapy + hormone therapy, 3 patients (5.7%) underwent chemotherapy + radiotherapy, 2 patients (3.8%) underwent chemotherapy and hormone therapy, and 1 patient (1.9%) was treated with surgery + interferon.

The mean follow-up time under psoriatic biologic therapy was 43.7 months (range: 19–81). All of the patients were followed up by the referring oncologist according to the type of diagnosed cancer, utilizing standardized follow-up methodologies and timelines specific to each cancer type.

Concerning survival and safety, 46 patients had regular follow-up. Six patients died—five from non-tumor-related causes—while the patient with endometrial cancer progressed during treatment and died from the tumor. Progression and death were considered as a natural progression of the patient’s tumor; no acceleration was attributed to the biologics, so the treatment with biologics was not interrupted. Most of them start biologic therapy as palliative care treatment in advanced and terminal oncologic disease.

Regarding drug discontinuation, the estimated biologic DS was 55.6% at 50 months (Figure 2).

## 4. Discussion

American and European guidelines lack data and recommendations specifically addressing the treatment of patients with concomitant psoriasis and a history of cancer. In European guidelines for the treatment of patients with cancer and psoriasis, strong recommendations are available for topical therapies, UVB-bs phototherapy, and acitretin, which are often insufficient for the treatment of severe psoriasis. There are also weak recommendations for systemic drugs such as apremilast and anti-TNFa, anti-IL 12/23, anti-IL23, and anti-IL17 agents on a case-by-case basis that includes discussion with an oncology specialist [4]. Therapies targeting anti-TNFa when used as a monotherapy in patients with moderate–severe psoriasis are not associated with an increased risk of solid tumor or lymphoreticular malignancy; thus, patients with a history of cancer may in some circumstances receive a TNFa inhibitor without expectation of an increased risk of tumor recurrence [5]. Anti-interleukin biologic agents have also been shown to have an excellent efficacy and safety profile in the general population, as well as in high-risk populations [6,7].

Evidence suggests that IL-17 is a key pathogenic factor involved in both early and late stages of tumorigenesis, and there is evidence of a lower risk of malignancies in subjects managed with IL-17 inhibitors. Experimental models have demonstrated that inhibition of IL-17 leads to attenuation of cancer development in a wide range of organs, including colon, pancreas, liver, skin, and lung. Preclinical cancer models have shown that inhibition of IL-17 suppresses metastasis and improves response to both chemotherapy and radiotherapy [8].

Similarly, several studies have directly and indirectly highlighted the potential role of IL-23 in modulating tumorigenesis in humans: IL-23 plays a role in suppressing natural or cytokine-induced innate immunity and promoting tumor development and metastases independently of IL-17A. Similarly, multiple lines of evidence suggest that IL-23 exerts a tumor-promoting effect. In particular, IL-23 can upregulate the growth and cell proliferation of oral squamous carcinoma [9], and intratumoral injection of IL-23-encoding mRNAs is associated with tumor regression in several cancer models [10]. 

Furthermore, serum levels of IL-17 and IL-23 in patients with pancreatic cancer are significantly higher than in the healthy population, and levels of IL-17 and IL-23 are significantly higher in those with stage III–IV tumors than in those with stage I–II tumors, suggesting that increased levels of IL-23/IL-17 in pancreatic tumor tissues may be a marker of prognosis [11]. 

In addition, the toll-like receptor 4 (TLR4) and IL-23/IL-17A axis plays an important role in tumor immunology: TLR4 is upregulated in HCC tissues, and the expression levels of IL-17A and IL-23, which are key mediators of inflammation that contribute to carcinogenesis, are correlated with TLR4 expression in HCC, showing that the this pathway may represent a new therapeutic target in HCC [12].

Given this large body of evidence, it is tempting to assume that targeting IL-17A and IL-23 might confer a protective role against tumor development in patients with moderate to severe psoriasis.

It has recently been shown that the balance between IL-12 and IL-23 is critical in carcinogenesis, since the T cell cytokines driven by IL-12 and IL-23 are particularly important in controlling tumor initiation, growth, and metastasis. IL-23 is known as an important cytokine that promotes tumorigenesis, and its levels correlate with poor prognosis in many human carcinomas. Therefore, the IL-12/IL-23 pathway may be a promising target for immunotherapy [13]. The overall safety profiles of ustekinumab remain favorable and did not show increased rates of malignancy in 3-year cumulative safety data from 3117 patients treated with ustekinumab across 4 studies [14].

DS in our cohort was around 80% at 24 months and 55.6% at 50 months. These data are in line with DS in the non-oncological psoriasis population treated at our center and from the data in the literature, showing that the drug was well tolerated over a long period of time in this special category of patients. Mastorino et al. reported on a case series of 1057 patients treated for psoriasis in our Turin Dermatology Clinic in which DS was 88% at 24 months for IL-23 inhibitors and 75% at 24 months for IL-17 inhibitors [15]. Similar percentages have been reported in other real-life data: in two studies by Torres et al., the cumulative probability of DS for all drugs was >79% at 18 months and >75% at 24 months for IL-23 inhibitors and IL-17 inhibitors [16,17].

The present study is an update of the paper published by Mastorino et al. in August 2022, which reported on a case series of 37 patients with a history of cancer treated for psoriasis with systemic biologic drugs at the Dermatology Clinic of Turin University Hospital [4]. Herein, we added 15 cases, resulting in a total of 52 oncology patients treated with biologic drugs for psoriasis at our clinic. Slightly more than half (28 of 52 patients, 54%) are taking an anti-IL-17 drug, while the remainder (24 of 52, 46%) are taking an anti-IL-23 agent. One patient died due to tumor progression, which was considered as a natural progression of disease that was independent of biologic treatment; however, no progression or recurrence of disease was reported in any of the other patients.

Similar case series in the literature are limited. In 2021, Bellinato et al. reported on 12 psoriatic patients with a previous or concomitant history of cancer treated with an anti-IL17 agent. Only two patients experienced progression, but these events were not related to the use of biologic drugs [18]. A study by Blauvelt et al. evaluated the cumulative safety experience with guselkumab, considering the incidence of adverse events such as cancer, using pooled data from phase-III VOYAGE 1 and 2 studies for up to 5 years, showing that the safety profile of this anti IL-23 drug remained constant and good over 5 years of continuous treatment [19].

Inference-based guidance was recently published by Papp et al., which aimed to explore the risk–benefit ratio in patients with a history of TST and psoriasis receiving systemic therapy, taking into consideration cancer prognosis, type and intensity of cancer treatment received, and patient preference. Papp showed that patients with a good prognosis of cancer will have similar outcomes to patients without TST, while patients with a poor cancer prognosis will still experience quality of life benefits from treatment for psoriasis that may outweigh the inherent theoretical risk of cancer death. Furthermore, there is no evidence of an increased risk of infection in cancer patients. The timing for starting treatment for psoriasis depends primarily on immune reconstitution and is thus based on the type of cancer treatment received rather than the type of tumor [20].

## 5. Conclusions

This is one of the largest case series in the literature on this subject. Data from this study confirm the current literature findings that do not suggest an increased risk of cancer, cancer recurrence, or disease progression in patients with a history of cancer and psoriasis treated with biologic agents. Biologic drugs represent a good option in the treatment of patients with a history of cancer, and this is a consistent case series of patients in a real-life setting supporting their use in this special population. IL-23 and IL-17 inhibitors, being immunomodulators instead of immunosuppressive drugs, may be a safe option to treat psoriasis in an active cancer setting and for moderate to severe immune-related adverse events. Larger cohorts with a longer follow-up and a prospective pharmacovigilance program to monitor carcinogenic risk are needed for all treatment categories.

## Figures and Tables

**Figure 1 jcm-13-06546-f001:**
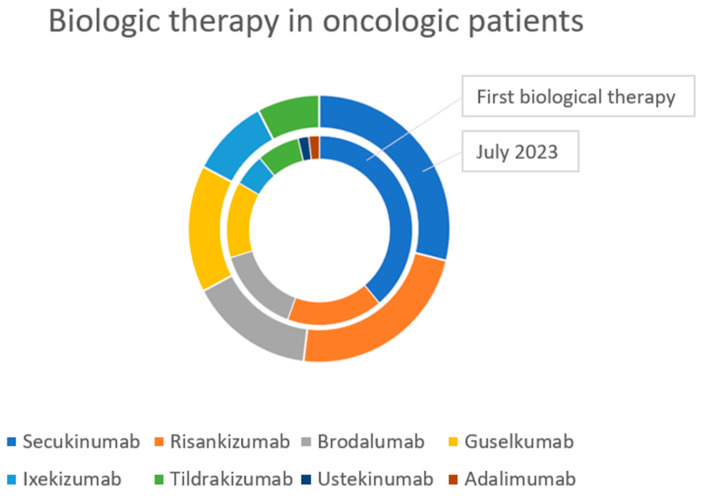
Biologic therapy in 52 oncologic patients with psoriasis.

**Figure 2 jcm-13-06546-f002:**
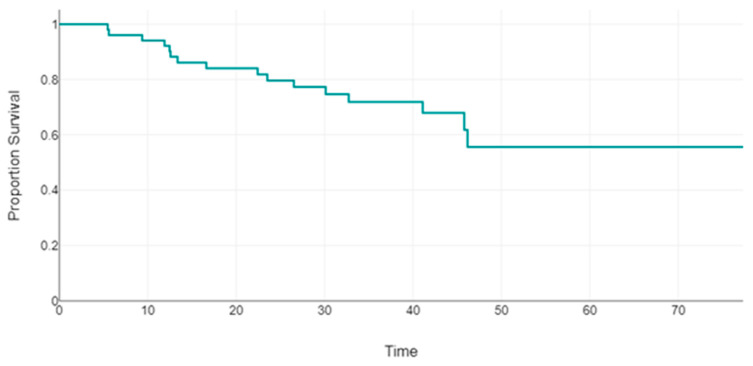
Patient survival (time in months) during biologic drug therapy (time and proportion survival).

**Table 1 jcm-13-06546-t001:** General characteristics of the population. PASI: Psoriasis Area Severity Index. DQLI: Dermatology Life Quality Index.

Characteristics of the Population	No. Patients (%)
Sex:	
- Male	31 (54%)
- Female	21 (46%)
Comorbidities:	
- Cardiovascular	26 (50%)
- Diabetes mellitus	9 (17%)
- Obesity	10 (19%)
Smoking:	
- Never smoked	24 (46%)
- Previously smoked	16 (31%)
- Actively smoked	12 (23%)
Age:	
- Median age	67 years (range: 40–90)
- Median age of onset of psoriasis	47 years (range: 4–80)
- Median age of cancer onset	55 years (range: 2–77)
Types:	
- Psoriasis vulgaris	41 (78%)
- Guttate psoriasis	4 (8%)
- Inverse psoriasis	4 (8%)
- Psoriatic erythroderma	2 (4%)
- Pustular psoriasis	1 (2%)
Difficult site involvement	42 (80%)
Psoriatic arthritis association	20 (38%)
Tumor type:	
- Gastrointestinal	7
- Melanoma	7
- Breast cancer	6
- Multiple non-melanoma skin cancer	6
- Bladder cancer	4
- Renal cancer	4
- Prostate cancer	4
- Bone cancer	3
- Meningioma	3
- Adrenal tumor	3
- Testicular cancer	2
- Pituitary tumor	2
- Endometrial cancer	1
- Ovarian cancer	1
- Leiomyosarcoma	1
- Kaposi sarcoma	1
- Acute lymphocyte leukemia	1
- Myeloproliferative disorder	1
- Neurinoma	1
Time of diagnosis:	
- Prior to the start of systemic therapy for psoriasis	49 (94%)
- During systemic therapy for psoriasis	3 (6%)
Median follow-up	43.7 months (range: 19–81)
PASI	
- At time zero	13.0
- At 52 weeks	1.1
DLQI	
- At time zero	22
- At 52 weeks	2.7

## Data Availability

The data are available from the corresponding author upon reasonable request.

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
