# Peer review of "Drug Survival and Clinical Course of Patients with Cancer Treated with Biologic Therapy for Psoriasis"

_jcm, 2024, doi:10.3390/jcm13216546_

Round 1

Reviewer 1 Report

Comments and Suggestions for Authors The aim of this article is: "to describe the characteristics of this population and to evaluate the efficacy, safety, and survival of the biological drug in this setting. “  Methodologically, it is a straightforward, descriptive, retrospective study. While one might suggest including a control group of patients without a history of cancer to assess drug survival comparatively, this was not the authors' objective.  The primary motivation for this study stems from concerns regarding the safety of biologic treatments for patients with both cancer and psoriasis. In their conclusion the authors state that: "Data from this  study confirm the current literature that does not suggest an increased risk of cancer, cancer recurrence, or disease progression in patients with a history of cancer and psoriasis treated with biological agents” However, in my opinion, it is difficult to draw such a conclusion without detailed information about the cancers involved, such as when and how they were treated, the length of remission, and other relevant details.  Publishing a study on the treatment of cancer patients with psoriasis without including these data seems inadequate.

Author Response

Thank you for your insightful comments. I would like to clarify that detailed information regarding the tumors involved, including treatment timelines, duration of remission, and other relevant data, is included in the text at line 142. Additionally, regular follow-up was managed by the referring oncology specialists to ensure comprehensive care for the patients.

Reviewer 2 Report

Comments and Suggestions for Authors

All comments are in atached file.

Author Response

  1. Line 64 - "Survival of the biological drug" has been replaced with "biological drug survival in the treatment of cancer patients with concomitant psoriasis."
  2. The reference has been added to the text.
  3. Linea 37 - Brief paragraph has been added to better explain biologic therapy "

    Biologics, or biological drugs, are targeted therapies derived from living organisms that modulate the immune system. They can be categorized based on their specific mechanisms of action in anti-TNF-alpha, anti-IL 23, anti IL-12/23 and anti-IL-17. Anti-TNF-alpha agents, such as adalimumab and etanercept, inhibit tumor necrosis factor-alpha, a pro-inflammatory cytokine involved in the pathogenesis of psoriasis. Anti-IL-23 biologics, like risankizumab, guselkumab and tildrakizumab, block interleukin-23, a cytokine crucial to the inflammatory process of the disease. Additionally, ustekinumab also targets both interleukin-12 and interleukin-23, helping to modulate immune responses. Anti-IL-17 therapies, including Secukinumab, brodalumab, ixekizumab and bimekizumab, inhibit interleukin-17, a cytokine that promotes inflammation and keratinocyte proliferation. Each class of biologics offers a unique mechanism of action, providing tailored treatment options for patients with moderate to severe psoriasis."

  4.  No
  5. The description has been moved below the figure, and "with psoriasis" has been added.
  6. The description has been moved below the figure, and the analysis has been extended to over 50 months, as shown in the graph.
  7. We have changed the sentence at line 155 to "American and European guidelines lack data and recommendations specifically addressing the treatment of patients with concomitant psoriasis and a history of cancer. "
  8. We have formatted the references according to the requested style.